# Integrated Analysis of the Transcriptome and Metabolome of *Brassica rapa* Revealed Regulatory Mechanism under Heat Stress

**DOI:** 10.3390/ijms241813993

**Published:** 2023-09-12

**Authors:** Jing Yu, Pengli Li, Song Tu, Ningxiao Feng, Liying Chang, Qingliang Niu

**Affiliations:** School of Agriculture and Biology, Shanghai Jiao Tong University, Shanghai 200240, China; 019150210006@sjtu.edu.cn (J.Y.); lipengli@sjtu.edu.cn (P.L.); 82170922@sjtu.edu.cn (S.T.); fnxalice@sjtu.edu.cn (N.F.) changly@sjtu.edu.cn (L.C.)

**Keywords:** heat stress, transcriptome, metabolome, brassica rapa

## Abstract

Affected by global warming; heat stress is the main limiting factor for crop growth and development. *Brassica rapa* prefers cool weather, and heat stress has a significant negative impact on its growth, development, and metabolism. Understanding the regulatory patterns of heat–resistant and heat–sensitive varieties under heat stress can help deepen understanding of plant heat tolerance mechanisms. In this study, an integrative analysis of transcriptome and metabolome was performed on the heat–tolerant (‘*WYM*’) and heat–sensitive (‘*AJH*’) lines of *Brassica rapa* to reveal the regulatory networks correlated to heat tolerance and to identify key regulatory genes. Heat stress was applied to two *Brassica rapa* cultivars, and the leaves were analyzed at the transcriptional and metabolic levels. The results suggest that the heat shock protein (HSP) family, plant hormone transduction, chlorophyll degradation, photosynthetic pathway, and reactive oxygen species (ROS) metabolism play an outstanding role in the adaptation mechanism of plant heat tolerance. Our discovery lays the foundation for future breeding of horticultural crops for heat resistance.

## 1. Introduction

*Brassica rapa chinensis* (*Brassica rapa*) is one of the most important leafy vegetables from the genus Brassica and is widely cultivated in Asia [1]. Global climate change has resulted in rising temperatures that threaten crop production. Currently, high temperature is the main limiting factor affecting plant growth and development. Heat stress often reduces photosynthetic efficiency and induces membrane dysfunction and oxidative damage in plants. *Brassica rapa* is sensitive to high temperatures, with a suitable growth temperature of 12–25 °C. When cultivated in high–temperature environments, the plant becomes more susceptible to infectious disease, leading to yellowing leaves, decreased quality, and an inferior yield [2]. To achieve an annual supply of *Brassica rapa* and develop heat–resistant varieties, it will be critical to understand the plant’s heat–resistance mechanisms and discover key heat–resistance genes.

As an adaptation to high–temperature environments in summer, plants have evolved various complex mechanisms to increase their heat resistance. The study of plant heat tolerance has become a hot topic in recent years. Plants initiate multiple physiological and molecular responses to enhance heat tolerance [3,4]. On the one hand, plants recruit antioxidant enzymes to eliminate reactive oxygen species; increase the transpiration rate to cool down; and increase soluble sugars, proteins, and proline to maintain cell membrane homeostasis [5,6]. On the other hand, plants sense via sensors and transduce heat stress signals to transcriptional regulators. Ca^2+^ and reactive oxygen species (ROS) are the initial signals that evoke a heat stress response (HSR) [7]. In this way, heat shock transcription factor (*HSF*) and *HSP* genes can be induced. It is well known that HSFs and HSPs play key roles in the induction of thermotolerance [8]. Plant–hormone–mediated pathways enhance plant heat tolerance by regulating gene expression. In addition, auxin, abscisic acid (ABA), ethylene (ET), and salicylic acid (SA) are reported to positively regulate thermal reactions [9]. In recent years, with the rapid development of high–throughput technology, transcriptome sequencing technology has been applied to the stress response research of many horticultural crops [10].

Recently, there have been some advances in the regulation mechanisms of plant heat stress in model plants. Due to the difficulties of genetic transformation in *Brassica rapa*, exploring the plant’s heat tolerance pathway has become extremely difficult. With next–generation sequencing technology, the process of identifying heat tolerance genes in *Brassica rapa* can be accelerated. The differences in heat stress responses to heat stress between heat–tolerant and heat–sensitive lines of *Brassica rapa* remain valuable for studying the mechanisms of plant heat tolerance. Based on our previous research on screening heat–resistant varieties of *Brassica rapa* [11], we used heat–tolerant ‘*WYM*’ and the heat–sensitive lines ‘*AJH*’ as test materials for short–term heat stress, analyzed their transcription and metabolic profiles, and determined the key transcription and metabolic pathways underlying the heat acclimation mechanisms of *Brassica rapa*.

## 2. Result

### 2.1. Plant Growth and Physiological Activities of Brassica rapa under Heat Stress

The ‘*WYM*’ line had a deeper leaf color, a narrower phenotype, and thicker stems compared to the ‘*AJH*’ line. Under high–temperature treatment, the leaves of the ‘*AJH*’ line were dehydrated and turned yellow, while those of the ‘*WYM*’ line showed normal performance. Seven days later, a large number of leaves of ‘*AJH*’ plants withered, some seedlings showed severe dehydration, and the leaves of the ‘*WYM*’ line faded with severe dehydration but still maintained a good growth trend (Figure 1). The physiological activities of *Brassica rapa* under heat stress are shown in Figure 2. The results indicated that ‘*AJH*’ was more sensitive to heat stress than ‘*WYM*.’ The important antioxidant enzyme activities were also measured. The activities of CAT, APX, POD, and SOD were increased in the ‘*AJH*’ and ‘*WYM*’ lines under heat stress compared to the activities under normal conditions. The POD and APX activities significantly increased under heat stress in the ‘*WYM*’ line. Under heat stress, the activities of these four enzymes were higher in ‘*WYM*’ than in ‘*AJH*.’ MDA content is a key indicator used for evaluating the degree of membrane oxidation. The high–level accumulation of ROS and malondialdehyde (MDA) in heat–sensitive cultivars resulted in a greater severity of damage to the photosynthetic apparatus and membrane system.

The level of MDA was higher in the ‘*AJH*’ line than that in ‘*WYM*’ line. Soluble sugars and proteins are important osmoregulation substances that play an important role in plant adaptations to heat stress. Recent studies have shown that soluble sugars regulate the ROS metabolic pathway. The contents of soluble protein and soluble sugar in ‘*WYM*’ leaves were found to be higher than those in ‘*AJH*,’ with 38.78% and 25.51%, respectively.

These results demonstrate that heat stress can affect plant physiology in many ways. Previous studies suggested that heat stress accelerates the degradation of chlorophyll a and b in leaves. Under heat stress, the contents of chlorophyll a and chlorophyll b were lower in the ‘*AJH*’ line than those in the ‘*WYM*’ line (Figure 1).

### 2.2. Transcriptome Profiles of Brassica rapa in Response to Heat Stress

To determine the transcriptome responses of *Brassica rapa* with different thermotolerance, four groups of cDNA libraries were constructed from the leaves of two *Brassica rapa* lines (‘*AJH*’ vs. ‘*WYM*’) under heat stress and normal conditions. In total, 76.78 Gb of clean data were obtained. The clean data of each sample totaled above 6.05 Gb in size, and the base percentage of Q30 was above 93.05%. For sequence alignment of the clean reads of each sample with the designated reference genome, the alignment rate ranged from 89.09% to 90.5%. Based on the quantitative expression results, a differential gene (DEG) analysis of the two groups was performed. In total, 7892 and 4455 DEGs were obtained from the ‘*AJH’* and ‘*WYM*’ groups, respectively. The corresponding Venn diagram shows that 1879 DEGs were co–expressed in the two groups. Specifically, 6013 and 2576 DEGs were expressed in the ‘*AJH*’ and ‘*WYM*’ lines (Figure 3). This result indicates that more gene expression changes occurred in heat–sensitive varieties, rather than heat–tolerant lines. Principal component analysis (PCA) was used to reduce the complexity of the data and allow us to more deeply explore the relationship between the samples and variation. Figure 3 shows that the ‘*AJH*’ and ‘*WYM*’ lines were separated in the PC1 dimension. The heat map of the clustering analysis suggests that the gene expression patterns in the ‘*AJH*’ line were relatively consistent with those of the control under heat stress. The box plot indicates that the gene expression distribution in the four sets of samples was basically consistent.

### 2.3. COG, GO and KEGG Pathway Analysis of DEGs 

A cluster of orthologous groups (COG) enrichment analysis was performed to annotate the functions of unknown sequences using known proteins. In the cellular processes and signaling category, posttranslational modification, protein turnover, and chaperone were the most commonly represented functional clusters in the two groups. Translation was also enriched (Figure 4). To further understand the functions of these gene sets, a Gene Ontology (GO) enrichment analysis was conducted. All differential expressed genes (DEGs) were divided into three parts: molecular function, cellular components, and biological processes. Figure 4 indicates that the highly enriched terms were the metabolic process, biological regulation, and cell parts in the ‘*WYM*’ and ‘*AJH’* lines. The catalytic activity term was also enriched in its molecular functions.

To gain a more detailed understanding of the metabolic pathways of *Brassica rapa* with different heat sensitivities under heat stress, a Kyoto Encyclopedia of Genes and Genomes (KEGG) pathway enrichment analysis was conducted. Histograms of KEGG showed that protein processing in the endoplasmic reticulum term and arginine and proline metabolism terms were most significantly enriched in the ‘*AJH*’ line. Many heat shock proteins have been discovered in this pathway and are involved in the responses of plants to heat stress. Proline (Pro) is one such component of plant proteins, whose content significantly increases under drought, salt, heat stress, cold stress, and freezing stress, while the ribosome and photosynthesis terms were prominent in the ‘*WYM*’ line. The ribosome is considered to be molecular machinery for protein synthesis. Plants adapt to heat stress by regulating photosynthesis under high temperatures. Therefore, we concluded that the differences in metabolic pathways between heat–tolerant and heat–sensitive lines determined the heat sensitivity of plants with good photosynthetic characteristics and that high ribosomal activity can effectively improve heat tolerance.

### 2.4. qRT–PCR Verification of RNA–Seq

To verify whether the transcriptome data were accurate, real–time quantitative PCR was performed to detect changes in the expression levels of candidate genes. HSP function as molecular chaperones. We selected several HSP genes from the significantly enriched pathways. The results showed that the expression trends of candidate genes were consistent with the transcriptome data (Figure 5).

### 2.5. Metabolic Profiles Analysis of Brassica rapa in Response to Heat Stress

Metabolomics analysis provides valuable information for characterizing heat tolerance in plants, and the composition and dynamics of the metabolome can be identified using metabolic profiles [12,13,14,15,16]. In this study, the metabolites of the ‘*AJH*’ and ‘*WYM*’ groups under heat stress and room temperature conditions were statistically analyzed, and a total of 440 and 377 metabolites were identified, respectively (Figure 6). Based on the expression of metabolites between different samples, a correlation heat map analysis and PCA principal component analysis were conducted to evaluate the similarity of samples within the group and the differences in samples between groups. PCA analysis of the metabolic profiling in the positive and negative ion modes is shown in Figure 6. The first and second principal components accounted for 32.44% and 14.73%, respectively, of the total variance. This result demonstrated that in ‘*AJH*,’ the metabolite expression patterns of ‘*WYM*’ varied significantly before and after heat stress. Partial least–squares discriminant analysis (PLS–DA) indicated that the separation degree of ‘*WYM*’ was greater, indicating a greater classification effect in ‘*WYM*’ compared to ‘*AJH*.’ The heat map illustrates the good correlation between four types of samples (*AJH*1 vs. *AJH*2 and *WYM*1 vs. *WYM*2).

### 2.6. Metabolic KEGG Pathways Analysis 

Based on the biological functional hierarchy of the metabolites, KEGG compounds were classified into several categories: compounds with biological roles, bioactive peptides, endocrine–disrupting compounds, pesticides, phytochemical compounds, and lipids. As shown in Figure 7, phospholipid, monosaccharide, and aminoglycoside terms were significantly enriched in ‘*WYM*,’ and carboxylic acids were dominant in ‘*AJH*.’ Phospholipids are the main components of cell membranes and play an important role in maintaining cell homeostasis. The present results indicated that phospholipids are important metabolites in the heat adaptation processes of *Brassica rapa*. Monosaccharides and aminoglycides are also important metabolites. Sugar metabolism is a key transcriptional and metabolic component used for distinguishing heat tolerance of the floral organs in rice [17]. Therefore, sugar metabolism also has a significant impact on plant heat stress responses. Metabolite difference analysis suggested the presence of organic acids and derivatives, lipids and lipid–like molecules, organic oxygen compounds, and organoheterocyclic compounds in ‘*WYM*’ and ‘*AJH*.’ Overall, organoheterocyclic compounds accounted for a large proportion of the total in ‘*WYM*,’ while organic acids and derivatives were the main metabolites in ‘*AJH*.’ To compare the differences in metabolite enrichment pathways between two varieties under heat stress and room temperature, KEGG pathway analysis was used. This analysis showed that cofactor biosynthesis was significantly co–enriched in these two lines. Flavone and flavanol biosynthesis and phenylpropanoid biosynthesis were also represented in all KEGG categories. Flavonoids are important antioxidants that can effectively eliminate oxygen–free radicals in plants. A previous study reported that phenylpropanoids and flavonoids play roles in the salinity tolerance of foxtail millet [18].

In summary, although there are differences in metabolic compounds between heat–resistant and heat–sensitive lines, these lines share certain pathways in the process of thermal adaptation, in which cofactors and flavonoids are important metabolites in *Brassica rapa* under heat stress.

## 3. Discussion

Since the response of plants to heat stress is a complex process, a combination of different techniques can help reveal the corresponding mechanisms at different levels. Under the dramatic climate changes currently happening worldwide, investigations into heat–tolerant mechanisms and crop improvements are important for global food security. Today, RNA sequencing (RNA–Seq) has become a powerful tool for exploring the mechanisms involved in abiotic stress and has been widely applied to numerous horticultural crops. Metabolites such as sugars, lipids, amino acids, organic acids, and nucleotides are essential in crops responding to abiotic stresses [19]. Metabolites are usually end products of complicated biochemical cascades associated with genomes, transcriptomes, and the phenotype of the proteome [20]. Current research on plant heat tolerance mainly focuses on physiological responses and transcriptional changes in plants, while the application of metabolomics to the study of heat tolerance in horticultural crops remains relatively rare. Metabolomics can be used to quantitatively analyze all metabolites in an organism and find the relative relationships between metabolites and physiological and pathological changes. In this study, integrated analysis of transcriptomics and metabolomics was applied for the first time to *Brassica rapa*. We also discussed the research progress of other researchers on heat stress in *Brassica rapa*, seeking to explore the mechanisms of heat adaptations in this crop more systematically and offering new directions for analyzing heat tolerance in horticulture research. 

### 3.1. The Role of Heat Shock Protein Family in Response to Heat Stress

Abiotic stress has an important impact on the normal growth of crops. Abiotic stress can change the conformation of cell proteins, cause the aggregation of unnatural proteins, and destroy the structure of cell membranes. HSPs are a type of stress protein synthesized by prokaryotes and eukaryotes under abiotic stress. As a molecular chaperone, the HSP is responsible for protein folding, assembly, translocation, and degradation in cells, maintaining the natural conformation of new polypeptides or recombining denatured proteins into natural conformation, thereby playing an important role in stabilizing the protein structure and maintaining cell homeostasis [8]. HSP70 is the most abundant protein produced in response to heat stress and acts as a molecular chaperone to protect protein folding and degradation, as well as improve stress tolerance [21,22]. In the endoplasmic reticulum processing term, we found that 12 *HSP70* genes were upregulated, with one downregulated. Three *HSP40* and four *HSP90* genes were upregulated in two lines. This result indicates that *HSP70* may play a pivotal role in the regulation of heat stress. 

Heat shock protein 20 (HSP20) is a type of functional protein that is widely synthesized by plants; HSP20 plays an important role in response to abiotic stresses such as drought, salt, and low temperatures and participates in the process of plant stress resistance [23]. In this study, 19 *HSP20* genes were upregulated in the ‘*WYM*’ and ‘*AJH*’ groups, and one *HSP20* gene was downregulated. Heat shock transcription factors (HSFs) play a crucial role in the responses of plants to abiotic stress [24]. A recent study demonstrated that *HsfA* is the main regulator of heat–stress–induced gene expression. Knockout of *HsfA2* in plants resulted in a much higher concentration of ROS. HsfA2 plays important roles in heat–induced oxidative damage. In this work, 11 HSPA1 genes were found in the KEGG pathway. Seven genes upregulated in the ‘*WYM*’ and ‘*AJH*’ lines and four genes downregulated in the ‘*WYM*’ line were upregulated in the ‘*AJH*’ line. Xie et al. reported that HsfA1a confers pollen thermotolerance by upregulating antioxidant capacity, protein repair, and degradation in *Solanum lycopersicum L.* [25]. Our data suggest that *HSP70*, *HSP20*, and *HsfA1*genes may play pivotal roles in heat stress survival in *Brassica rapa*.

### 3.2. Plant Hormone Transduction Signaling in Response to Heat Stress 

Plant hormones are trace substances in plant metabolism that participate in all stages of plant growth and development and play an important role in stress adaptation. Auxin is an important regulatory factor in plant growth. Auxin is commonly present in various plant tissues and mainly involved in cell wall formation and nucleic acid metabolism. A variety of phytohormones take part in plant growth and development and safeguard stress responses [9]. Yue et al. found that protein processing in the endoplasmic reticulum and plant hormone signal transduction pathways could help *B. rapa* resist stress and regulate leaf senescence [2].

Recent studies have shown that *GH3* (green hagen 3), *3–Indoleacetic acid* (IAA), *auxin response factor* (ARF) and *small auxin up RNA* (SAUR) are the main genes that respond to early IAA and are the primary expression genes induced by auxin [26]. The *GH3* gene encodes an auxin–binding enzyme, and the *GH3* protein functions as an indoleacetic acid aminotransferase synthase, which has a feedback regulatory effect on auxin by reducing the level of free auxin. The *GH3* gene not only participates in the light signaling pathway but also acts in response to drought, high temperature, and other abiotic stresses by regulating plant growth to adapt to adverse environments. Aux/IAA protein plays an extremely important role in the auxin signaling pathway. Mutants of this pathway exhibit developmental phenotypic changes directly related to auxin. *ARFs* affect gene expression and are involved in auxin regulation. *ARFs* can also bind to auxin response elements. Thus, *IAA* and *ARF* genes play important roles in the IAA signaling pathway. Within plant hormone transduction signaling, 11 *IAA* genes were upregulated, and one was downregulated. Four *GH3* genes were upregulated, and one was downregulated. In addition, ten *SAUR* genes were downregulated, and one was upregulated. We also found that six *JASMONATE–JIM–DOMIN* (JAZ) genes were significantly downregulated in both varieties. JAZ is a repressor protein that functions in JA responses under stress conditions. In addition, two *JA–amino acid synthetase* (JAR1) genes were downregulated in the two lines. JA signal transduction pathways also participated in the process of cold acclimation in wheat. *Ethylene responsive factors* (ERFs) are important regulatory factors of the ET signaling pathway in stress responses. ET is known to reduce thermotolerance to heat stress by deterring antioxidant defenses [27]. *ERFs* also play important roles in numerous types of abiotic stress such as cold, drought, heat, salt, and freezing [28]. We found that two *ERF1* and two *ERF2* genes were downregulated in the ‘*WYM*’ and ‘AJH’ lines. Notably, ERF1 was downregulated by 10.38 and 5.44 in ‘*AJH*’ and ‘*WYM*,’ respectively. Thus, IAA, ET, and JA signaling played dominant roles in heat–stress–adaptation mechanisms. Interestingly, we found that two *PATHOGENESIS RELATED* (PR) genes and one *NON–EXPRESSOR OF PR GENE 1* (NPR1) were downregulated [29]. *NPR1* is an SA receptor involved in the SA–dependent activation of PR genes.

Compared with the metabolomic data, JA and SA were significantly downregulated in the plant hormone transduction pathway, while IAA and ERF signaling were upregulated in the plant hormone transduction pathway, which was consistent with the transcriptome results. Recently, Quan et al. reported that the thermotolerance of *Brassica rapa* in heat stress responses could be improved through exogenous glycine betaine (GB) [30]. In this study, ABA, SA, auxin, and cytokinin (CK) hormones were either up– or downregulated in GB–primed *Brassica rapa* plants under heat stress. This finding further supports our results, showing that ABA and SA signaling may be involved in the thermotolerance of *Brassica rapa* in heat stress responses. Notably, we observed that many genes were screened in the IAA signaling pathway responding to heat stress responses. The *ERF1* genes were significantly upregulated in both heat–sensitive and heat–resistant lines. The cross interaction of hormones in response to heat stress is a complex process, and it remains to be determined which signaling pathway dominates and how the various pathways cooperate with each other.

### 3.3. The Photosynthetic Pathway in Brassica rapa Is Enhanced under Heat Stress

Considerable evidence indicates that the photosynthetic process is highly sensitive to heat stress. The photoexcitation of chlorophyll (Chl) molecules embedded in the photosynthetic complex of thylakoid membranes has been observed, and light energy was used by Photosystem II (PSII) for the photooxidation of water. The released electrons moved from PSII via the transport chain (ETC) of Plastoquinone/Plastoquinone (PQ), cytochrome b6f (cytb6f), and plastid anthocyanidin (Pc) to Photosystem I (PSI). In this paper, we found significant enrichment in the photosynthesis and photosynthesis antenna protein pathways [31]. The light absorbed by the chlorophyll of Photosystem II and I facilitated aerobic photosynthesis. In the photosynthesis pathway, genes involved in photosystem II (*PsbP*, *PsbQ*, *PsbR*, *PsbW*, *PsbY*, *Psb27*, and *Psb28*) and photosystem I (*PsaE*, *PsaG*, *PsaH*, and *PsaK*) were regulated in response to heat stress in ‘*WYM*’ (Figure 8). These gene were all upregulated. The genes involved in photosynthetic electron transport (five P*etE*, six *PetF*, and one *PetH* genes) were also upregulated. The Photosystem contains two main components: a light trapping complex and a reaction center complex. These complexes play a key role in photoprotection by dissipating excessive absorbed light energy through induction and regulation [32]. In higher plants, the main light harvesting complex is trimeric LHCII. LHCII is the most abundant membrane protein, accounting for about 30% of thylakoid membrane protein and half of the total Chl [33]. LHCII is composed of homologous heterotrimers of three protein subtypes: LHCB1, LHCB2, and LHCB3. 

Research has shown that the absence of *LHCB1* gene leads to chlorophyll loss, a light–green phenotype, and delayed growth in plants [34]. *LHCB1* encodes a component of the light acquisition complex LHCII in Photosystem II. In the photosynthesis–antenna protein pathway, five *LHCB1* genes, one *LHCB2* gene, one *LHCB4* gene, and one *LHCB5* gene were significantly downregulated in both the ‘*WYM*’ and ‘*AJH*’ lines from the transcriptome (Figure 8). Therefore, *LHCB* genes responded to heat stress by altering the light acquisition complex LHCII in Photosystem II. Interestingly Zhang et al. found that five *LHCB* genes were downregulated compared to the control in two *Brassica rapa* lines under heat stress, while the expression levels were higher in heat–tolerant lines compared to those in heat–sensitive lines [1]. Even during the recovery period, *LHCB1* gene expression remained at a low level, indicating that photosynthetic devices suffered damage and that the damage was more severe in heat–sensitive lines under heat stress.

Metabolomic data indicated that the chlorophyll degradation pathway was also significantly enriched. A proteomics study indicated that 258 heat–responsive proteins in wheat play roles in redox regulation, chlorophyll synthesis, protein turnover, and carbon fixation [35]. We investigated the porphyrin metabolism pathway in the ‘*WYM*’ and ‘*AJH*’ lines under heat stress. The results suggested L–Glycine was upregulated in the ‘*WYM*’ and ‘*AJH*’ lines, while protochlorophyllide, sedoheptulose 7–phosphate, porphobilinogen, coproporphyrin I, urobilinogen, and pyropheophorbide a were downregulated in the ‘*AJH*’ line. These results indicate that the chl degradation process of heat–sensitive lines was more frequent, which is consistent with previous experimental results. Moreover, the abundance of proteins involved in photosynthesis increased in heat–tolerant wheat under heat stress [36].

In the pathway of carbon fixation between light and organisms, we found that D–Sedoheptulose 7–phosphate was downregulated in the ‘*WYM*’ line, whereas malic acid and sedoheptulose were upregulated. In the ‘*AJH*’ line, D–Sedohe ptulose 7–phosphate was downregulated under heat stress compared with the control. Malic acid and sedoheptulose are products of carbon fixation. D–Sedoheptulose 7–photosphate is also an important product of photosynthesis and the pentose phosphate pathway. The heat–tolerant lines may have a stronger impact on carbon fixation. Heat stress also caused a reduction in the light–absorbing efficiency of both photosystems (PSI and PSII). The Fv/Fm ratio is known to be an important indicator of the maximum quantum efficiency of PSII. For instance, Sharma et al. successfully obtained heat–tolerant and heat–sensitive types of wheat from 1274 wheat cultivars under heat stress using Fv/Fm phenotyping [37]. To verify the inhibitory effects of heat stress on the photosynthesis of sensitive and heat–resistant *Brassica rapa*, we measured the photosynthetic rate, electron transport rate, and maximum photochemical efficiency of PS II (Fv/Fm) (Figure 9). The results showed that the photosynthetic rate, electron transport rate, and maximum photochemical efficiency of *Brassica rapa* significantly decreased under heat stress, with heat–sensitive varieties having a stronger inhibitory effect under heat stress. The photosynthetic pathway, photosynthetic antenna protein pathway, and chlorophyll degradation pathway may be important metabolic pathways that affect the sensitivity of *Brassica rapa* to heat stress.

### 3.4. ROS Metabolic Pathways Are Altered under Heat Stress in Brassica rapa

Heat stress provokes the production of a large amount of ROS. In tomatoes, heat stress mainly activates the expression of proteins that ensure protein quality and ROS detoxification [38]. Plants can restore redox homeostasis and cellular damage via ROS scavenging mechanisms. ROS–scavenger enzyme–related genes were identified from the DEGs. In the reactive oxygen metabolism pathway, we found significant upregulation of the *superoxide dismutase* (SOD1), *Catalase* (CAT), *ascorbate peroxidase* (APX), and *ascorbic acid oxidase* (AAO) genes. *CAT* and *SOD* genes were especially highly expressed in ‘*WYM*.’ SOD is considered to be the first line of defense in plant cells against ROS toxicity. APX enzyme also participates in the heat stress resistance of plants. These results were verified in previous experiments. To investigate the scavenging effects of antioxidant enzymes on ROS, the contents of OH^–^ and O^2–^ in the ‘*WYM*’ and ‘*AJH*’ lines under heat stress and normal conditions were measured. The contents of O^2–^ and H_2_O_2_ in the ‘*AJH*’ line were higher than those in ‘*WYM’* line. These results indicate that under heat stress, more ROS accumulate in heat–sensitive varieties, which will cause oxidative damage to plants, further injuring the membrane system and causing an imbalance in ROS metabolism (Figure 9). In another study, Zhang et al. also found that *BrSOD1* and *BrSOD2* were strongly expressed in heat–tolerant lines [1]. In addition, the authors found that *Peroxidase* (prx) and *glutathione peroxidase* (GPX1) genes were responsive to heat stress in *Brassica rapa*. However, the relevant physiological indicators were not validated in the previous study.

Although no significant upregulation of the *prx* genes was found in RNA–seq, the measured peroxidase (POD) enzyme activity was higher in the heat–resistant line than that in the heat–sensitive line, further confirming previous research. Due to differences in variety selection, the types of significantly upregulated antioxidant enzyme genes were also different. In short, the proposed expression regulation among these antioxidant enzyme genes also makes important contributions to the resistance of *Brassica rapa* to heat stress.

In summary, significant differences were investigated in transcription and metabolic pathways between the heat–tolerant and heat–sensitive lines in *Brassica rapa*. Notably, the heat shock protein family and multiple pathways involved in ROS metabolism, porphyrin degradation, plant hormone transduction, and photosynthesis play important roles in the heat sensitivity of *Brassica rapa*. In addition, the CRISPR/Cas genome editing method can help researchers achieve accurate, robust, and efficient genetic manipulation in plant genomes, especially to engineer the abiotic stress tolerance of horticultural crops [39,40]. The effects of candidate genes responsible for heat tolerance on the levels of metabolites and transcripts could be further validated with the new–emerging CRISPR/Cas 9 technique. Thus, more efforts are required in the future to study the heat stress responses of crops.

## 4. Material and Methods

### 4.1. Plant Cultivation and Heat Stress Treatment

Two *Brassica rapa* varieties (heat–tolerant ‘*WYM*’ line and heat–sensitive ‘*AJH*’ line) were grown in a controlled phytotron of Shanghai Jiao Tong University, Shanghai, China. The full and uniform seeds were presoaked in distilled water and placed on moistened filter paper overnight to promote germination. Sprouting seeds were placed in trays containing a 3:1 mixture of soil and vermiculite. The environmental conditions were set as 25 ± 2 °C with a photoperiod of 16 h light and 8 h dark, 150 mol·m^−2^·s^−1^ light intensity, and a relative humidity of 55%. High–temperature treatment (42 °C 6 h) was applied to the ‘*WYM*’ line and ‘*AJH*’ line when they developed five fully opened leaves, and another group was placed in a room temperature environment as a control. The third function leaves were collected from four groups of *Brassica rapa* plants after 6 h of heat stress treatment for quantitative RT–PCR. The leaf samples were immediately quick–frozen in liquid nitrogen and then stored in an ultra–low temperature refrigerator at –80 °C for RNA–seq. Three biological replicates were applied for the transcriptome analysis.

### 4.2. Sample Collection and RNA Sequencing

The data were analyzed on the online platform of Majorbio Cloud Platform (www.majorbio.com) (accessed on 30 May 2023). Total RNA was extracted from the leaves of *Brassica rapa* seedlings. Total RNA was extracted using RNA Extraction Kit (Tiangen, Beijing, China) following the manufacturer’s protocol. The concentration and purity of the extracted RNA was detected by Nanodrop2000, and agarose gel electrophoresis was used to detect the integrity of RNA, which is shown in Appendix A. The mRNA was enriched using Oligo dT, fragmented reverse–transcribed into anti cDNA, and linked to adapter. The library was constructed, and three independent replicate RNA samples were used for each library. Then, target bands were collected, qualified by TBS380 (Picogrreen), and mixed according to the data ratio on the machine. Finally, the target bands were sequenced on an Illumina platform, and paired–end reads were generated. Clean data were obtained, which were compared with the reference genome to obtain mapped data (reads) for subsequent analysis. At the same time, the quality of the alignment results was evaluated, and sequence alignment analysis was performed using TopHat 2 software. The overall quality evaluation was conducted, including the sequencing saturation, sequencing coverage, distribution of reads in different regions, and distribution of reads in different chromosomes.

### 4.3. Differential Expression Gene Analysis

The read counts of each sample gene were obtained using the results of the comparison to the genome and the genome annotation file. Then, the fragments per kilobases per million reads (FPKM) were estimated and normalized to obtain the standard.

### 4.4. Go and KEGG Pathway Analysis

Based on the *B. rapa* V3.0 genome (http://brassicadb.cn/#/Download/ (accessed on 3 June 2023) [41], the Cufflinks software (http://coletrap/nelllab.github.io/cufflinks/ ) (accessed on 3 June 2023) was used to assemble and splice mapped reads, which were compared with known transcripts. All DEGs were annotated through the Gene Ontology (GO) and Kyoto Encyclopedia of Genes and Genomes (KEGG) databases to determine their biological functions and participation pathways. GO enrichment analysis was performed on the genes using the Goatools software (https://github.com/tanghaibao/GOatools) (accessed on 3 June 2023) [42]. The KEGG pathway enrichment analysis was performed using the R package.

### 4.5. Real Time Quantitative PCR

RNA samples were extracted using the RNAprep pure plant kit (Tiangen, Beijing, China), and cDNA synthesis was performed using the PrimeScript™ RT reagent Kit (Takara, Japan). Quantitative real–time PCR was performed using SYBR Prime^TM^ Script (TakaRa, Japan). The quantitative Real–Time PCR (qRT–PCR) reaction mixture included 2 µL template cDNA, 0.4 µM each of the forward and reverse primers, 7.2 mL dd H_2_O, and 10 µL SYBR PrimeScript RT enzyme Mix. PCR amplification was conducted with 60 °C as the annealing temperature using step one plus real–time PCR system (Applied Biosystems, Carlsbad, CA, USA). The melting curve is shown in Appendix A. Each analyzed gene was tested with three biological replicates and three technical replicates. The *Actin gene* (*Bra028615*) was used as an internal control, and the 2^−ΔΔCT^ method was used to calculate relative gene expression levels [43,44]. Specific primers were designed by Becon Designer 7.9 software. The primer sequences used in the qRT–PCR analysis are shown in Appendix A. The Specific primers were aligned with blast in the Brassica database (http://brassicadb.cn) (accessed on 3 June 2023) to confirm its specificity.

### 4.6. Statistical Analyses

Statistical analyses were performed using Microsoft Office Excel 2016 software (Redmond, WA, USA). Analysis of variance (ANOVA) was completed with SPSS 25.0 software. Statistically significant differences between means were determined at *p* < 0.05 using Tukey’s test.

### 4.7. Physiological Index Measurement

The MDA, CAT, POD, and SOD activities were measured by test kits (Comin, Suzhou, China) according to the manufacturer’s instructions [45]. Three independent biological replicates were applied for each treatment. The soluble sugars were measured by a previously described method [46]. The soluble protein was measured as described in a previous study [47]. The relative conductivity was quantitated as previously described [38]. The chlorophyll content was determined according to the method described in [48].

### 4.8. Metabolomic Analysis

A total of 5 mg freeze–dried leaf samples were extracted with 400 µL a mixture of 80% methanol and water under sonication for 30 min at low temperature (5 °C). Then, the suspension was centrifuged for 15 min at 13,000 rpm. The ProtID Chip–43 II column supernatant was transferred for metabolic profiling analysis. Sample extracts were injected (600 ng) into an UHPLC–Q Exactive HF–X. The molecules were separated through a ACQUITY UPLC HSS T3 column (100 mm × 2.1 mm). The mobile phases consisted of H_2_O with 0.1% formic acid (FA) as solution A and acetonitrile (ACN) with 0.1% FA as solution B.

### 4.9. Data Processing and Differential Metabolites Identification 

Principal component analysis (PCA) was conducted to detect the intrinsic variation between the ‘*AJH*’ leaves and the ‘*WYM*’ leaves. Orthogonal partial–least squares discriminant analysis (OPLS–DA) was used to effectively distinguish the ‘*AJH*’ leaves and the ‘*WYM*’ leaves. Differential metabolites were screened as those with a variable influence on projection (VIP) value greater than 1 and a *p* value less than 0.05. Based on mass spectrometry identification, all metabolites were compared with the KEGG and HMDB databases to obtain annotation information of metabolites in the database, and their annotation status in the database was statistically analyzed.

## 5. Conclusions

Crops have evolved physiological, metabolic, and molecular regulatory mechanisms to react and adapt to heat stress. Analysis of heat stress is of great importance due to the increasingly frequent and severe occurrence of high–temperature stress. To relieve the damage of heat stress on crops, we must investigate, using multiple tools, the mechanisms by which plants respond to heat stress. In this study, the physiological index and transcriptome and metabolome analyses were used to decipher the molecular mechanisms in *Brassica rapa*. The results suggest that the HSP, plant hormone transduction, photosynthetic pathway (the light–harvesting chlorophyll a/b binding (LHC) gene family), and ROS catabolism signaling play leading roles in the heat tolerance of the two lines.

## Figures and Tables

**Figure 1 ijms-24-13993-f001:**
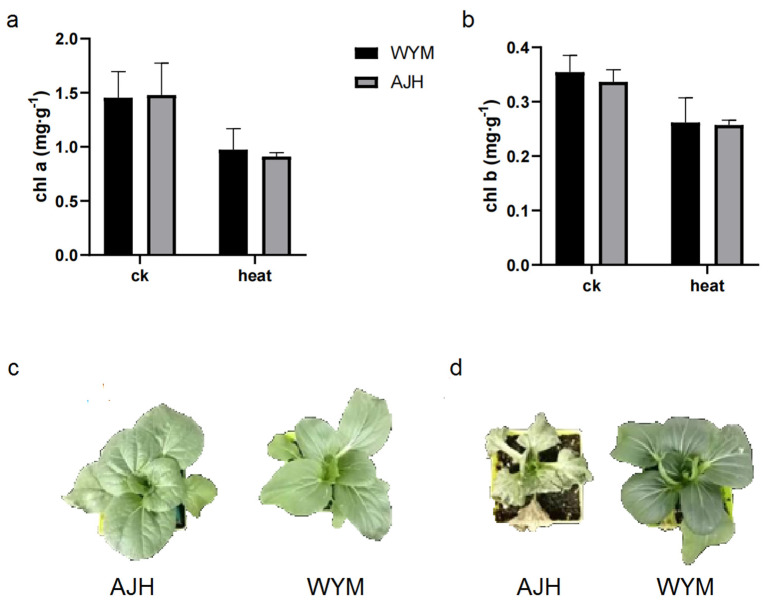
(**a**,**b**) The chlorophyll a and chlorophyll b content under heat stress and normal conditions. (**c**,**d**) The phenotype of the ‘*AJH*’ and ‘*WYM*’ lines under heat stress and normal conditions.

**Figure 2 ijms-24-13993-f002:**
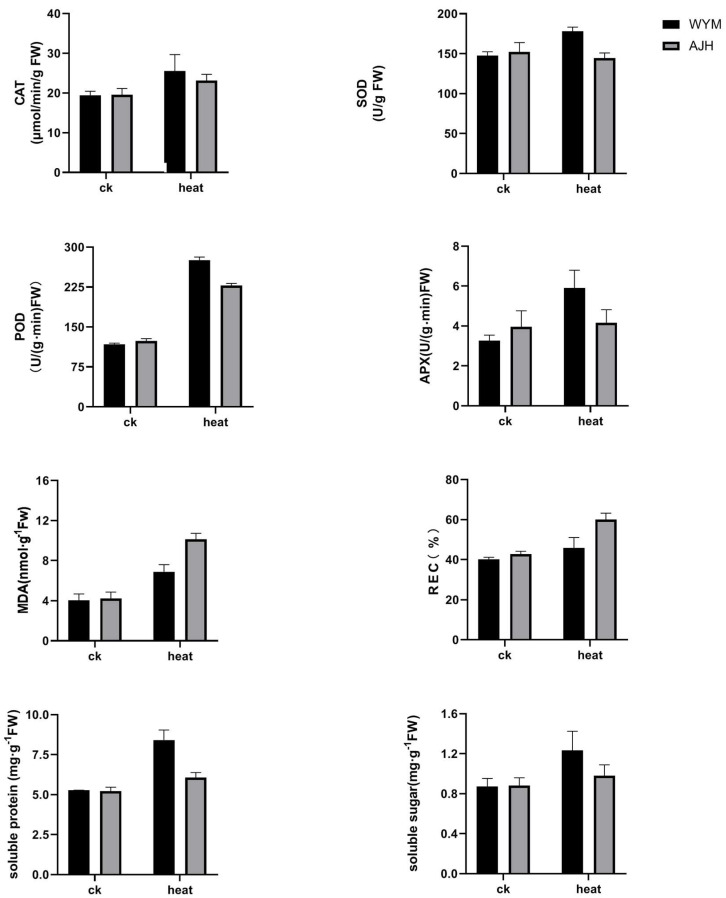
Determination of physiological indices of *Brassica rapa* under heat stress and normal conditions. The activities of four significant antioxidant enzymes, relative electrical conductivity (REC), and the contents of MDA, soluble sugar, and soluble protein were detected. Data are the means (±SD) of three independent experiments. Error bars denote the standard deviation.

**Figure 3 ijms-24-13993-f003:**
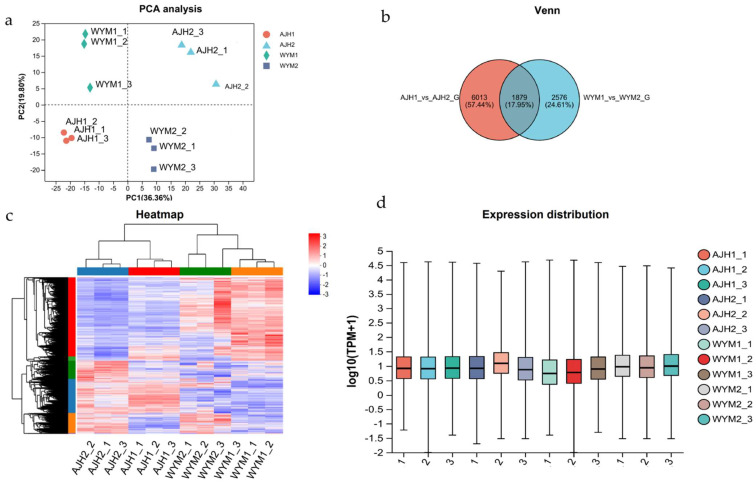
(**a**) Principal component analysis (PCA) of the RNA–Seq data. (**b**) Numbers of DEGs in each comparison between the heat stress treatment and control. (**c**) Cluster dendrogram of different samples. The color scale (blue to purple) indicates correlation between samples. (**d**) Box plot of expression distribution in ‘*AJH*’ and ‘*WYM*’ line under heat stress and normal condition. Each color in the figure represents a sample, and the horizontal line in the figure represents the median gene expression in the sample.

**Figure 4 ijms-24-13993-f004:**
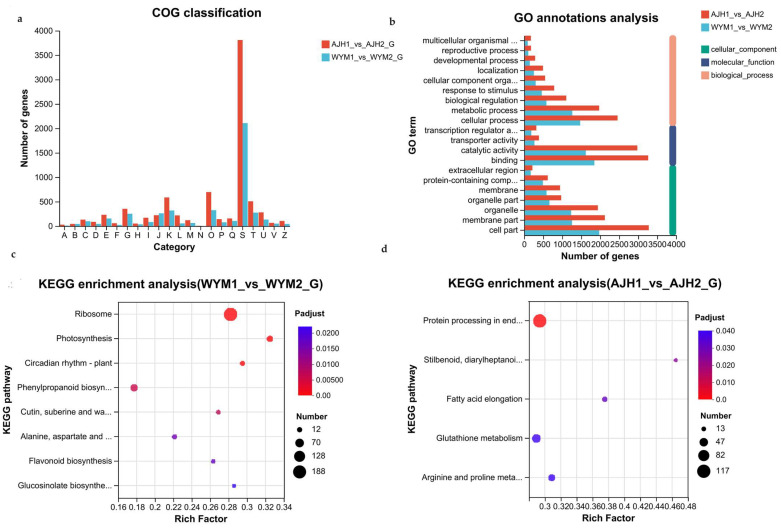
Identification and functional annotation of DEGs. (**a**) COG functional classification; (**b**) histogram of GO classification; (**c**,**d**) KEGG pathway enrichment analysis of DEGs in the ‘*AJH*’ and ‘*WYM*’ lines. The vertical axis represents the name of the pathway, and the horizontal axis represents the ratio of the number of genes/transcripts enriched in the pathway to the number of annotated genes/transcripts (background number).

**Figure 5 ijms-24-13993-f005:**
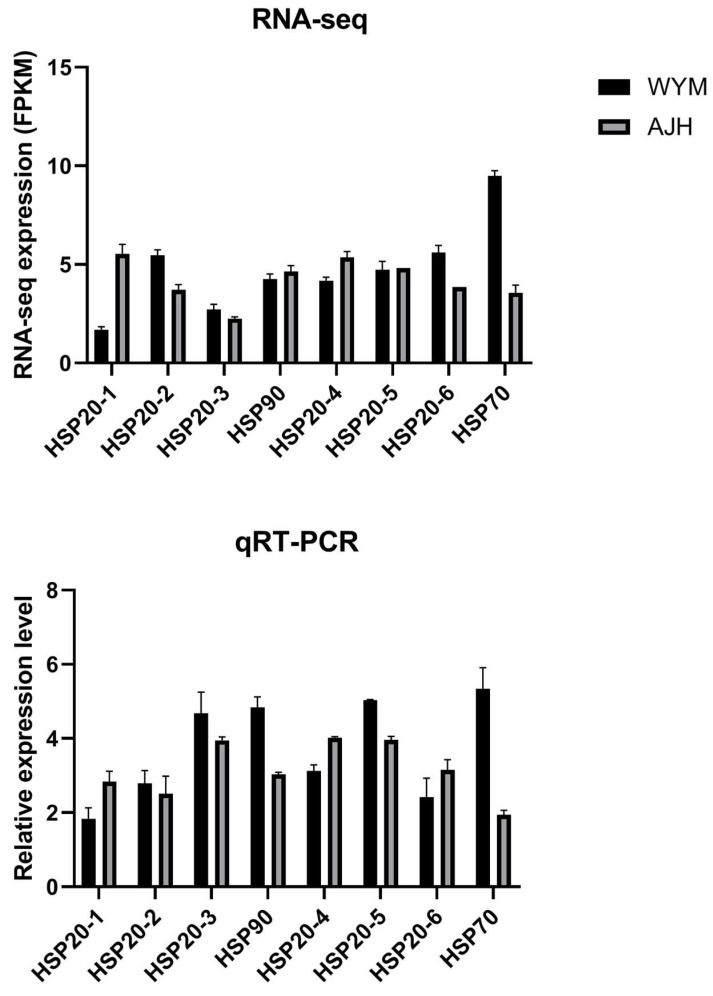
Expression profiles of eight candidate genes (HSPs) by qRT–PCR and RNA–seq. The expression analysis of candidate genes in two *Brassica rapa* accessions under normal and heat stress conditions. Vertical bars indicate the STDEV of each treatment. Each treatment included three plants.

**Figure 6 ijms-24-13993-f006:**
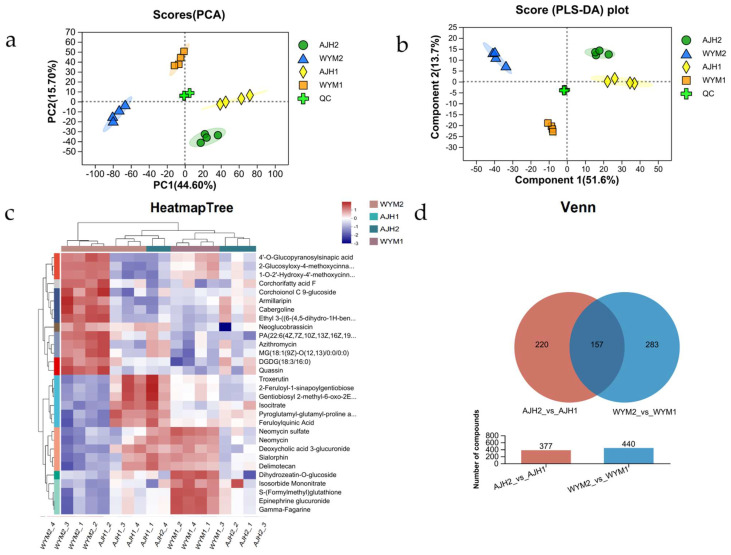
Differential metabolite analysis on *Brassica rapa* (‘*WYM*’ vs. ‘*AJH*’) lines. (**a**) Principal component analysis displaying 8 samples along the axes of PC1 and PC2; PCA analysis displaying 4 samples along the axes of PC1 and PC2, which describe 32.44% and 14.73% variability, respectively; (**b**) partial least–squares discriminant analysis; (**c**) heat map of correlation between samples; (**d**) Venn diagram of differential metabolites *Brassica rapa* (‘*WYM*’ vs. ‘*AJH*’) lines under heat stress and normal conditions.

**Figure 7 ijms-24-13993-f007:**
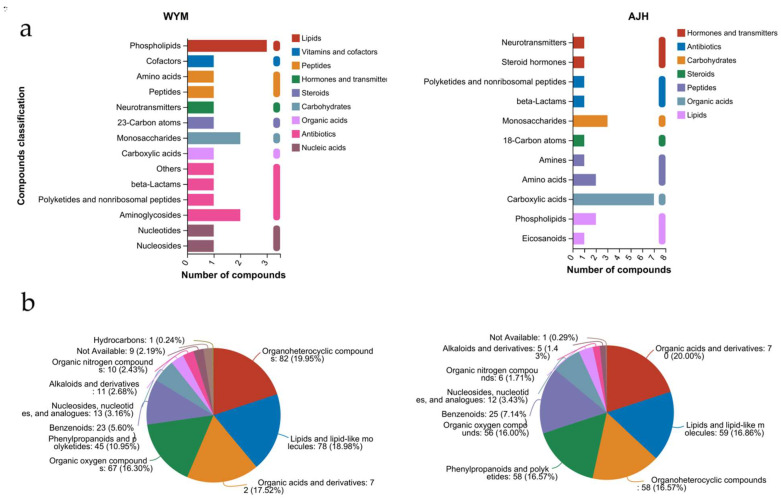
Metabolic KEGG enrichment analysis of the ‘*AJH*’ and ‘*WYM*’ groups under heat stress and normal conditions. (**a**) KEGG compound classification; (**b**) compound classification of the human metabolome database (HMDB); (**c**) KEGG enrichment analysis. * indicates significant difference (*p* < 0.05), ** indicates a highly significant difference (*p* < 0.01). *** indicates a highly significant difference (*p* < 0.001).

**Figure 8 ijms-24-13993-f008:**
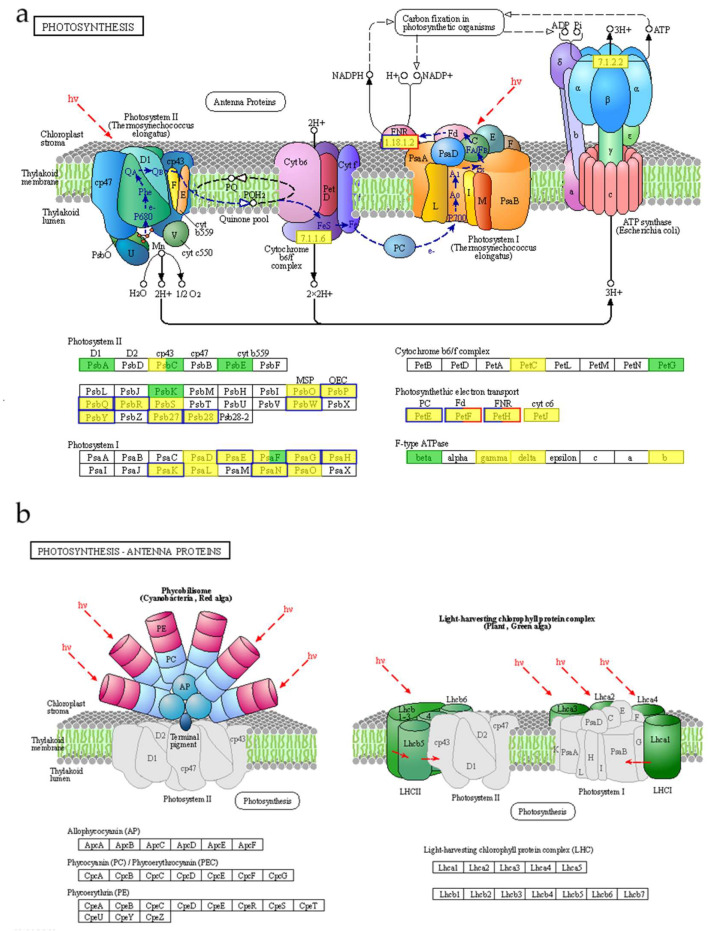
The diagram of the photosynthesis pathway. (**a**) The photosynthesis pathway and antenna protein pathway. (**b**) The photosynthesis–antenna protein pathway.

**Figure 9 ijms-24-13993-f009:**
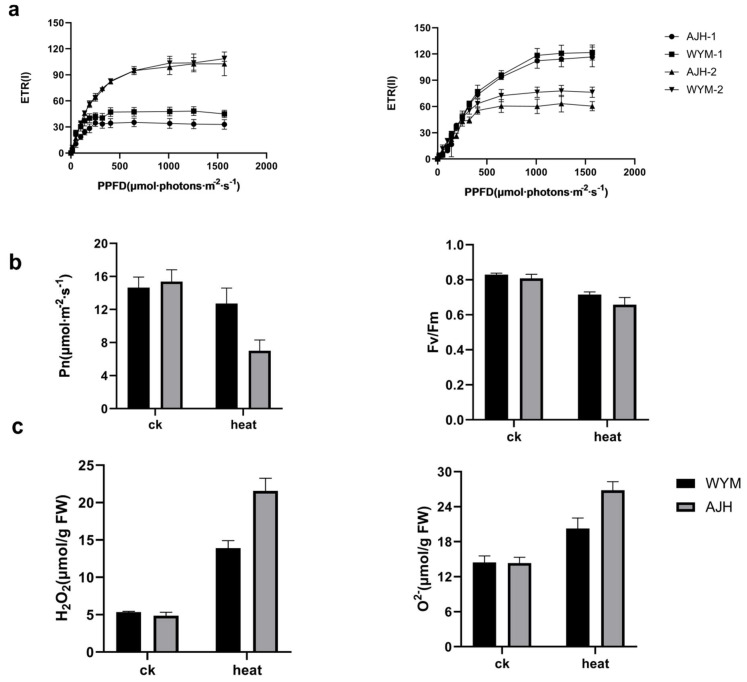
(**a**) Rapid light curve (RLC) of the photosynthetic electron transport rate of PSII and PSI (ETRII and ETRI) of *Brassica rapa* leaves; (**b**) the Fv/Fm and net photosynthetic rate (Pn) of the two *Brassica rapa* lines under heat stress and normal conditions. (**c**) analysis of O^2–^ and H_2_O_2_ content of the two *Brassica rapa* lines under heat stress and normal conditions. Each treatment included three plants. Data are the means (±SD) of three independent experiments.

## Data Availability

The data presented in this study are available in this manuscript.

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
