# Peer review of "Integrated Analysis of the Transcriptome and Metabolome of Brassica rapa Revealed Regulatory Mechanism under Heat Stress"

_ijms, 2023, doi:10.3390/ijms241813993_

Round 1

Reviewer 1 Report

The manuscript could it be published as it stands. However, I would recommend the authors to carefully read again the manuscript and correct some mistakes reagarding the articles referenced. Some references do not match with what the authors report.

I am very glad the authors wrote this manuscript. It is a well-written, needed, and useful . The authors are analytical and the tables/shcematics that they are presenting indeed help the author very much. I belive that the manuscript could be published prior to some minor revision. And after the authors have checked again some minor typos that exist. Also the fonds of some figures should be increased in size.

Research questions are well defined and within the aims and the scope of the journal.   The introduction is adequate and includes in suitable way the relevant earlier publications.   Materials are almost properly described.   Methods are also almost properly described and used in a way that is possible to replicate.   The investigation is performed to good technical standards.   It is no ethical problem involved.   A nicely conducted research (although a bit complicated) with conclusions well supported by the results.   The level of English is adequate.    Moreover, the selection of the plant species test needs to justified.   Also please use uniform letter fonts.   An increase font size in figures 6 and 7.

Author Response

Dear reviewers:

Thank you very much for your comments concerning our manuscript, these comments are very valuable and helpful for revising and improving our paper, as well as the important guiding significance to our research. We have studied comments carefully and have made correlation which we hope meet with approval. Revised portion are marked in red in the paper. The responds to reviewer’s to the comments are as followings:

Comment 1: I would recommend the authors to carefully read again the manuscript and correct some mistakes regarding the articles referenced. Some references do not match with what the authors report.

Response: We apologize for making mistakes in the reference. We have correct, added and revised the reference according to the publisher's requirements. The added literature includes:

Li, X., Lawas, L.M., Malo, R., Glaubitz, U., Erban, A., Mauleon, R., Heuer, S., Zuther, E., Kopka, J., Hincha, D.K., Jagadish, K.S. Metabolic and transcriptomic signatures of rice floral organs reveal sugar starvation as a factor in reproductive failure under heat and drought stress. Plant Cell Environ. 2015, Oct;38(10):2171-92.

Pan, J., Li, Z., Dai, S., Ding, H., Wang, Q., Li, X., Ding, G., Wang, P., Guan, Y., Liu, W. Integrative analyses of transcriptomics and metabolomics upon seed germination of foxtail millet in response to salinity. Sci Rep. 2020, Aug 12;10(1):13660.

Yu, J., Cheng, Y., Feng, K., Ruan, M., Ye, Q., Wang, R., Li, Z., Zhou, G., Yao, Z., Yang, Y., Wan, H. Genome-wide identification and expression profiling of tomato Hsp20 gene family in response to biotic and abiotic stresses. Front Plant Sci. 2016, Aug 17;7:1215.

Scharf, K.D., Berberich, T., Ebersberger, I., Nover, L. The plant heat stress transcription factor (Hsf) family: Structure, function and evolution. Biochim. Biophys. Acta. 2012, 1819, 104–119

Xie, D.L., Huang, H.M., Zhou, C.Y., Liu, C.X., Kanwar, M.K., Qi, Z.Y., Zhou, J. HsfA1a confers pollen thermotolerance through upregulating antioxidant capacity, protein repair, and degradation in Solanum lycopersicum L. Hortic Res. 2022, Apr 13;9:uhac163. doi: 10.1093/hr/uhac163. PMID: 36204210; PMCID: PMC9531336.

Verma, V., Ravindran, P., Kumar, P.P. Plant hormone-mediated regulation of stress responses. BMC Plant Biol. 2016,16:86.

Goldental-Cohen, S., Israeli, A., Ori, N., Yasuor, H. Auxin response dynamics during wild-type and entire flower development in tomato. Plant Cell Physiol. 2017, Oct 1;58(10):1661-1672.

Munné-Bosch, S., Peñuelas, J., Asensio, D., Llusià, J. Airborne ethylene may alter antioxidant protection and reduce tolerance of holm oak to heat and drought stress. Plant Physiol. 2004, Oct;136(2):2937-47; discussion 3002.

Achard, P., Gong, F., Cheminant, S., Alioua, M., Hedden, P., Genschik, P. The cold-inducible CBF1 factor-dependent signaling pathway modulates the accumulation of the growth-repressing DELLA proteins via its effect on gibberellin metabolism. Plant Cell. 2008, 20(8):2117-29.

Loake, G., Grant, M. Salicylic acid in plant defence-the players and protagonists. Curr Opin Plant Biol. 2007,10(5):466-72

Quan, J., Li, X., Li, Z., Wu, M., Zhu, B., Hong, S.-B., Shi, J., Zhu, Z., Xu, L., Zang, Y. Transcriptomic analysis of heat stress response in Brassica rapa L. ssp. pekinensis with improved thermotolerance through exogenous Glycine Betaine. Int. J. Mol. Sci. 2023, 24, 6429.

Sattari Vayghan, H., Nawrocki, W.J., Schiphorst, C., Tolleter, D., Hu, C., Douet, V., Glauser, G., Finazzi, G., Croce, R., Wientjes, E., Longoni, F. Photosynthetic light harvesting and thylakoid organization in a CRISPR/Cas9 Arabidopsis Thaliana LHCB1 knockout mutant. Front Plant Sci. 2022, Mar 7;13:833032.

 Mazor, Y., Borovikova, A., Caspy, I., Nelson, N. Structure of the plant photosystem I supercomplex at 2.6 Šresolution. Nat Plants. 2017, Mar 1;3:17014.

Peter, G.F., Thornber, J.P. Biochemical composition and organization of higher plant photosystem II light-harvesting pigment-proteins. J Biol Chem. 1991, Sep 5;266(25):16745-54.

 Lu, Y., Li, R., Wang, R., Wang, X., Zheng, W., Sun, Q., Tong, S., Dai, S., Xu, S. Comparative proteomic analysis of flag leaves reveals new insight into wheat heat adaptation. Front Plant Sci. 2017, Jun 20;8:1086.

Wang, X., Dinler, B.S., Vignjevic, M., Jacobsen, S., and Wollenweber, B. Physiological and proteome studies of responses to heat stress during grain filling in contrasting wheat cultivars. Plant Sci. 2015, 230, 33-50.

 Ruban, A.V., Wentworth, M., Yakushevska, A.E., Andersson, J., Lee, P.J., Keegstra, W., Dekker, J.P., Boekema, E.J., Jansson, S., Horton, P. Plants lacking the main light-harvesting complex retain photosystem II macro-organization. Nature. 2003, Feb 6;42 1(6923):648-52.

 Sharma, D.K., Andersen, S.B., Ottosen, C.O., Rosenqvist, E. Phenotyping of wheat cultivars for heat tolerance using chlorophyll a fluorescence. Funct. Plant Biol. 2012, 39, 936-947. doi: 10.1071/FP12100

Mazzeo, M.F., Cacace, G., Iovieno, P., Massarelli, I., Grillo, S., and Siciliano, R.A. Response mechanisms induced by exposure to high temperature in anthers from thermo-tolerant and thermo-sensitive tomato plants: a proteomic perspective. PLoS One. 2018,13:e0201027.

 Klap, C., Yeshayahou, E., Bolger, A.M., Arazi, T., Gupta, S.K., Shabtai, S., Usadel, B., Salts, Y., Barg, R. Tomato facultative parthenocarpy results from SlAGAMOUS-LIKE 6 loss of function. Plant Biotechnol J. 2017, May;15(5):634-647.

Comment: The selection of the plant species test needs to justified. Also please use uniform letter fonts.

Response: Thank you to the reviewer for their valuable suggestions. We conducted measurements of physiological indicators, including the content of soluble sugar, soluble protein and MDA ,the antioxidant enzyme activity, and Fv/Fm value. These data can explain the heat resistance of Brassica rapa, and we have added literature to illustrate the heat resistance of the variety. In addition, we corrected the letter fonts.

Comment 2:An increase font size in figures 6 and 7.

Response: We apologize for the difficulty caused by unclear images for readers. We have resized the Figure 3, 4,6,7, in the hope of a clearer representation.

Thank you again for the reviewers' affirmation and suggestions on this paper. We have also polished the language of the paper to meet the publishing requirements.

Reviewer 2 Report

The study was focused on integrated analysis of the transcriptome and metabolome of Brassica rapa revealed regulatory mechanism under heat stress. Heat stress was applied to two Brassica rapa cultivars, and the leaves were examined at the transcriptional and metabolic levels. The Authors revealed that HSP family, plant hormone transduction, chlorophyll degradation, photosynthetic pathway, and ROS metabolism play an outstanding role in the adaptation mechanism of plant heat tolerance.

I noticed several points that need to be corrected in the manuscript:

-        There is the lack of Table S1 regarding primer sequences used in the qRT-PCR analysis.

-        I recommend including the electropherograms presenting the RNA bands in agarose gels in the manuscript or in the Supplementary file – it would provide information regarding quality of total RNA samples. The RNA concentration should be quantified at both 260 and 280 nm, and purity of RNA should be calculated based on the A260/A280 coefficient.

-        If Authors used SYBR Green fluorescent dye during RT-PCR gene expression studies, hence, it is obligatory to perform Melting Curve Analysis, and results of this examination should be added in the manuscript or Supplementary file (e.g., JPG or TIFF file).

-        The number of significant and relevant citations is too low.

-        Discussion part should be significantly deepened, and extended. At present, the discussion is plain/superficial.

-        There are several typing errors in the manuscript

-        Extensive editing of English language is required.

In my opinion, the manuscript should be rejected, considerably improved, and than re-submitted.

-        Extensive editing of English language is required.

Author Response

Dear reviewers:

Thank you very much for your comments concerning our manuscript. These comments are very valuable and have greatly improved the quality of our paper. We tried our best to improve our manuscript and made some changes in the manuscript. These changes will not influence the content and framework of the paper. Revised part were marked in red in the paper. We have responded point-to-point to the reviewer's comments.

Comment 1:There is the lack of Table S1 regarding primer sequences used in the qRT-PCR analysis.

Response: We are very sorry for not attaching the Table S2 due to our negligence. We added it in the supplementary materials.

Table S2 The Specific primers sequences used for qRT-PCR

Gene ID

forward primer

reverse primer

BraA06g001970.3C

TTCAAGGAGTTATGGTTA

TAGGAATCACAACATTAAG

BraA08g014770.3C

AATGTGTTCAATCTTCTC

GAATATACTGTAGGTTGTTAT

BraA03g027160.3C

GATGAAGTGAAGATAGAG

TATCAACATTATCAGGTAG

BraA03g009000.3C

ATCATCAACACCTTCTAC

ATACAACAACAACAGTAATC

BraA04g010550.3C

TTCTTAACCTAAACAACATTCCAT

CTTCCTTCTTCAGTCCTTCT

BraA03g011180.3C

ACAAGAACATACTTCAGAT

CAACTTAAACCTCCTCAT

BraA03g005120.3C

TGTGAAGTATGTGAGGAT

GTTACAAACAGCAGAGAT

BraA10g018140.3C

ATAAGAAGGATGTTAGTG

AAGAATCAATCTCAATAGT

Comment 2: recommend including the electropherograms presenting the RNA bands in agarose gels in the manuscript or in the Supplementary file -it would provide information regarding quality of total RNA samples. The RNA concentration should be quantified at both 260 and 280 nm, and purity of RNA should be calculated based on the A260/A280 coefficient.

Response: As suggested by the reviewers, we have supplemented relevant information concerning RNA used for experiment, including electropherograms, concentrations and OD260/280.

Comment 3: If Authors used SYBR Green fluorescent dye during RT-PCR gene expression studies, hence, it is obligatory to perform Melting Curve Analysis, and results of this examination should be added in the manuscript or Supplementary file (e.g., JPG or TIFF file).

Response: it is real true as reviewers suggested that we have provided pictures of the melting curve.

Comment 4: The number of significant and relevant citations is too low.

Response: Considering the comments of the reviewers, we have added some literature to support our conclusions, in order to make the paper more reliable. The added literature includes:

Comment 5:Discussion part should be significantly deepened, and extended. At present, the discussion is plain/superficial.

Response: Thank you for the valuable comments in terms of discussion. We have also added some discussion on relevant literature. We have read papers by other researchers on the heat tolerance mechanism of Brassica rapa, including transcriptome studies on using GB to enhance the heat tolerance of Brassica rapa, as well as transcriptome studies on different time slot of Brassica rapa under heat stress. Their research is very meaningful and supports our conclusion that the photosynthetic hormone pathway and ROS detoxification mechanism play an important role in the heat tolerance of Brassica rapa. The main discussion content added is as follows, detailed in the main text:

Today, RNA sequencing (RNA-Seq) has become a powerful tool for exploring the mechanisms involved in abiotic stress and has been widely applied to numerous horticultural crops. Metabolites such as sugars, lipids, amino acids, organic acids, and nucleotides are essential in crops responding to abiotic stresses[28]. Metabolites are usually end products of complicated biochemical cascades associated with genomes, transcriptomes, and the phenotype of the proteome[29]. Current research on plant heat tolerance mainly focuses on physiological responses and transcriptional changes in plants, while the application of metabolomics to the study of heat tolerance in horticultural crops remains relatively rare. Metabolomics can be used to quantitatively analyze all metabolites in an organism and find the relative relationships between metabolites and physiological and pathological changes. In this study, integrated analysis of transcriptomics and metabolomics was applied for the first time to Brassica rapa. We also discussed the research progress of other researchers on heat stress in Brassica rapa, seeking to explore the mechanisms of heat adaptations in this crop more systematically and offering new directions for analyzing heat tolerance in horticulture research.

Xie et al. reported that HsfA1a confers pollen thermotolerance by upregulating antioxidant capacity, protein repair, and degradation in Solanum lycopersicum L[34]. Our data suggest that HSP70, HSP20, and HsfA1 may play pivotal roles in heat-stress survival in Brassica rapa.

 A variety of phytohormones take part in plant growth and development and safeguard stress responses [35]. Yue et al. found that protein processing in the endoplasmic reticulum and plant hormone signal transduction pathways could help B. rapa resist stress and regulate leaf senescence.

Recently, Quan et al. reported that the thermotolerance of Brassica rapa in heat stress responses could be improved through exogenous glycine betaine [40]. In this study, abscisic acid (ABA), salicylic acid (SA), auxin, and cytokinin hormones were either up- or down-regulated in GB-primed Brassica rapa plants under heat stress. This finding further supports our results showing that ABA and SA signaling may be involved in the thermotolerance of Brassica rapa in heat stress responses. Notably, we observed that many genes were screened in the IAA signaling pathway responding to heat stress responses. The ERF1 genes were significantly up-regulated in both heat-sensitive and heat-resistant lines. The cross interaction of hormones in response to heat stress is a complex process, and it remains to be determined which signaling pathway dominates and how the various pathways cooperate with each other.

Interestingly Zhang et al. found that five LLHCB genes were down-regulated compared to the control in two Brassica rapa lines under heat stress, while the expression levels were higher in heat-tolerant lines compared to those in heat-sensitive lines[1]. Even during the recovery period, LHCB gene expression remained at a low level, indicating that photosynthetic devices suffered damage and that the damage was more severe in heat-sensitive lines under heat stress.

In another study, Zhang et al. also found that BrSOD1 and BrSOD2 were strongly expressed in heat-tolerant lines[1]. In addition, the authors found that prx (Peroxidase) and GPX1 genes were responsive to heat stress in Brassica rapa. However, the relevant physiological indicators were not validated in the previous study.

Although no significant upregulation of the prx genes was found in RNA-seq, the measured POD enzyme activity was higher in the heat-resistant line than that in the heat-sensitive line, further confirming previous research. Due to differences in variety selection, the types of significantly upregulated antioxidant enzyme genes were also different. In short, the proposed expression regulation among these antioxidant enzyme genes also makes important contributions to the resistance of Brassica rapa to heat stress.

Comment 6: There are several typing errors in the manuscript.

Response: I'm sorry for some incorrect writing in the article. We have corrected the writing errors.

Comment 7:Extensive editing of English language is required.

Response: We are Sorry for our poor drafting. We have polished the paper and improved the image to make it more understandable.

We appreciate for reviewers’ warming work earnestly, and hope the correlation will meet with approval.Once again, thank you very much for your comments and suggestions

Round 2

Reviewer 2 Report

The manuscript has been correctly revised. In my opinion, it may be considered for publication.